# Polariton condensation and surface enhanced Raman in spherical ZnO microcrystals

Victor V. Volkov[1], Daniel J. Oliver[1] & Carole C. Perry [1✉]

Preparation and characterization of polariton Bose–Einstein condensates in micro-cavities of high quality are at the frontier of contemporary solid state physics. Here, we report on three-dimensional polariton condensation and confinement in pseudo-spherical ZnO microcrystals. The boundary of micro-spherical ZnO resembles a stable cavity that enables sufficient coupling of radiation with material response. Exciting under tight focusing at the low frequency side of the bandgap, we detect efficiency and spectral nonlinear dependencies, as well as signatures of spatial delocalization of the excited states which are characteristics of dynamics in polariton droplets. Expansion of the photon component of the condensate boosts the leaky field beyond the boundary of the ZnO microcrystals. Using this, we observe surface polariton field enhanced Raman responses at the interface of ZnO microspheres. The results demonstrate how readily available spherical semiconductor microstructures facilitate engineering of polariton based electronic states and sensing elements for diagnostics at interfaces.

---

[1] Interdisciplinary Biomedical Research Centre, School of Science and Technology, Nottingham Trent University, Clifton Lane, Nottingham NG11 8NS, UK. ✉email: Carole.Perry@ntu.ac.uk

Since the paradigm of a cavity to provide effective coupling of an external field with a material's response[1], a range of devices based on microspheres[2], pillar micro-resonators[3], wires[4], and 1D micro-cavities[5] has inspired research in lasing dynamics[6] and in quantum correlation processes[7]. At the same time, the discovery of Bose–Einstein condensation (BEC) of polaritons has been reported in micro-cavities of high quality[8]— the discovery of which has yet to find practical application. Within this context, the properties of Zinc oxide (ZnO) are interesting: ZnO has an ultraviolet band gap (~3.37 eV), a large exciton binding energy (~60 meV), and a large oscillator strength[9]. Polariton lasing[10] and room temperature BEC have been observed in ZnO micro-cavities[11,12]. Further, whispering galleries due to one- and two-dimensional confinement of cavity modes have been reported for ZnO microstructures with high-quality surfaces[13–15]. Despite these successes, experimental realization of polariton confinement in three-dimensions remains challenging[16,17].

In our studies, we explore the potential for polariton BEC in two readily available representative ZnO microstructures of pseudo-spherical geometry, which provide conditions that resemble a stable cavity[18,19]. The first is ZnO analytical standard microcrystals (size ca. 100 μm) from Sigma-Aldrich, and the second are unannealed ZnO polycrystalline microspheres (particle size ca. 2 μm) that can be readily synthesized[20]: please, refer to Methods and Supplementary Information Note 1. Taking advantage of spherical shapes, which resemble a stable cavity, we report on polariton droplets in ZnO microstructures and the associated surface field enhancement mechanism. Specifically, under tight focussing of the resonant field in spherical microspheres we provide opportunities to stimulate polariton dynamics. Confocal emission microscopy for such states reports initially quadratic intensity power dependence when carriers are excited at the red side of the band-gap under a tight focusing regime. Above a threshold, the power dependence demonstrates saturation. Upon increase of the excitation power, images, (detected exciting the band-gap carriers) demonstrate a systematic blue shift. Both, nonlinear efficiency and the spectral dependence on fluence are specific to parametric generation of droplets and excitons[1,21–24].

Using Fourier transform spatial spectroscopy we separate images of excitons-polaritons experiencing delocalization and localizing tendencies. The former is characteristic of the condensation regime, while the latter is descriptive for relaxation mechanisms associated with dissipation via defects. At the high-power limit, the leaky field component of the polaritons provides the opportunity for surface field enhancement behaviour. Indeed, using the discovered regime, we observed surface enhanced Raman for GLHVMHKVAPPR (G12) polypeptide that we have previously used to investigate the thermodynamics of interaction with zinc oxide[25], and for Si–O⁻ stretching vibrations in $Q_3$-structural elements[26,27] of a borosilicate glass in contact with ZnO microspheres. The surface polariton field enhancement (SPFE) mechanism reported here opens up opportunities for research and engineering of bio-inorganic and inorganic composites using zinc oxide as both a structural and sensing element.

## Results

**Linear emission of ensembles.** Figures 1a, b are emission spectra for the two samples, detected under excitation at both 330 and 405 nm. For complete characterization of energy relaxation pathways, we use two-dimensional assemblies of emission spectra[28] using excitation from 240 to 600 nm, as shown in Fig. 1c, d. Both systems demonstrate prominent contributions of the band-gap excitons (centred at about 3.7 eV) and relatively weak "green" and "red" emissions at 510 and 650 nm, respectively, which are assigned to surface associated traps[9]. The observed spectral properties are typical for ZnO materials, when excited using relatively low (about 0.1 μW cm⁻²) power.

To address the nature of optical states in the spectral region of the band gap, in Fig. 1e, f we show excitation spectra (horizontal slices in Fig. 1c, d), detecting emission at 580 nm. The spectra of the two systems demonstrate inhomogeneous broadening at the red side of the edge emission. At room temperature, this is the spectral region where longitudinal optical phonons dominate in the emission response from ZnO bulk[9]. However, as a result of the spherical character of the two ZnO systems, here, we may also expect a contribution of the lower polariton branch[29] due to coupling between light and matter: in fact, a spherical cavity is expected to demonstrate the highest possible quality factor[30]. In this respect, it is encouraging that the spectral broadening at the red side of the edge emission is prominent in perfect ZnO microspheres (see Fig. 1d). Anticipating the nature of the observed spectral inhomogeneity in the emission excitation spectra, further,

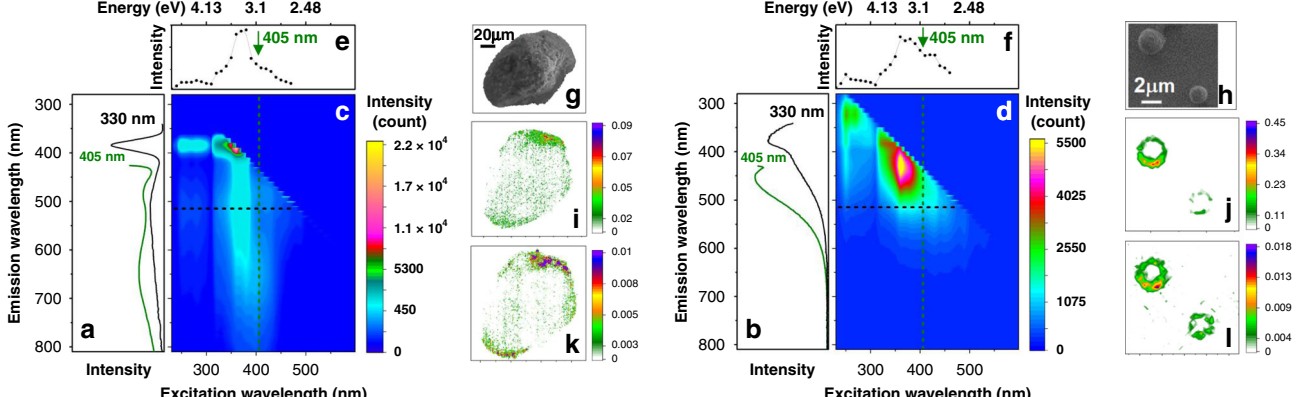

**Fig. 1 Emission properties of ZnO microcrystals in the bulk and using microscopic resolution. a, b** Emission spectra from analytical standard ZnO microcrystals and microspheres, respectively, in the bulk using excitation wavelengths of 330 nm and 405 nm. **c, d** Two-dimensional steady-state emission spectrum of standard microcrystals and microspheres, respectively. **e, f** Emission excitation spectrum for standard microcrystals and microspheres, correspondingly, detected for emission at 580 nm. Green arrows indicate excitation conditions used in confocal microscopy studies. **g, h** SEM image of a selected microcrystal (the scale bar is 20 μm) and selected microspheres (the scale bar is 2 μm), respectively. **i, j** Confocal microscopy images scanned at the equatorial plane of the microcrystal and the microspheres, respectively, detecting at 430 nm using 405 nm for excitation. **k, l** Confocal microscopy image scanned at the equatorial plane of the microcrystal and the microspheres, correspondingly, detecting at 510 nm using 405 nm for excitation.

in our microscopy studies, to stimulate polariton dynamics in the lower polariton branch, we use coherent radiation at 405 nm (Fig. 1a, b, green lines). At the same time, this wavelength avoids possible bistabilities, which have been anticipated for polaritons under intense excitation at the blue side of the band gap[31].

In Fig. 1g–l we relate SEM images of the selected microstructures with confocal microscopy scans at the equatorial plane detecting at 430 and 510 nm under excitation at 405 nm. In the case of the large microcrystal, emission specific to the states associated with the band-gap are nearly uniform. However, slight increases of the emission density are present at the edges of greater curvature, see Fig. 1i. The emission at 510 nm tends to localize at the surface, see Fig. 1k. Emission becomes strongly associated with the surface when we employ in-gap excitations. In Supplementary Note 2 we compare images as shown in Fig. 1i, k with such detected at 510 nm using 488 nm for excitation; also, we present radial distribution functions for emission intensities detected in microcrystals of different sizes under the described excitation and detection conditions. In the case of the smaller microspheres emission dominates at the edges of the structures. This provides the peculiar rim patterns as may be seen in emission images of small microspheres as shown in Fig. 1j, l. Comparing microscopy scans for such structures using emission at 430 and 510 nm, we observe only slight spatial broadening.

**Nonlinear emission of selected microcrystals**. To discover the nature of the carriers, in Fig. 2a we present the intensity of emissions at 430 nm sampled in commercial microcrystals (line with open circles) and in the synthesized microspheres (line with closed circles) in dependence on fluence (power per area in the beam waist) of the excitation field at 405 nm (at the low-frequency side of the band gap transition). Both samples demonstrate quadratic dependence for the fluence below $2 \times 10^5$ W cm$^{-2}$. The quadratic dependence is steeper for the small microspheres. To explain the efficiency dependencies at both low and high-power limits, we explore spectral properties in

dependence on power of the exciting field at 405 nm: Fig. 2c, d. For convenience, we normalize the spectra on the maxima. Upon excitation power increase, the spectral response in a selected ZnO microsphere demonstrates a dramatic enhancement and narrowing of the spectral feature at about 460 nm, initially. It is striking that with further power increase the emission shifts to higher frequency: see Fig. 2d. In the case of the spectral response in a large microcrystal, we observe similar tendencies, see Fig. 2c. Additionally, in the latter system, for fluence above $2.9 \cdot 10^5$ W cm$^{-2}$, spectra demonstrate relatively narrow spectral modulations: see star marks in Fig. 2c. Whispering gallery (WG) modes in ZnO microspheres excited above the band-gap have previously been modelled using eigenvalue formalism in a symmetric finite-element method[20]; here, we use finite difference time-domain (FDTD) theory for the experimental geometry and describe properties of gaussian beam focusing under a scalar approximation. We simulate a diversity of cavity modes and their properties in dependence on polarization and position of excitation for complete ZnO microspheres with radii from 0.6 to 1.8 µm. The results of simulations suggest that the peculiar rim patterns in the emission images of small microspheres (Fig. 1j, l) is mainly due to the WG modes stimulated along the excitation radiation axis: for details, please refer to Supplementary Notes 3–8. Simulations confirm the Q-factor for the UV modes to vary between 70 and 140 for the range of sizes. Accordingly, provided with a proper fluence[14], we may expect stimulation of the polariton regime in 1–2 µm radius ZnO spheres, as observed in high quality cavities[1,8]. Furthermore, theory instructs that a smaller cavity would sustain a lower number of spectrally broader cavity modes with higher intensity at the higher frequency side that occupy a larger relative fraction of the internal volume close to the surface. Under comparable fluence, smaller spheres would experience a higher density of energy in a smaller number of cavity modes. As a result of such confinement, in smaller microspheres, we may expect stronger scattering and discord in polariton dynamics stimulated in the UV spectral domain.

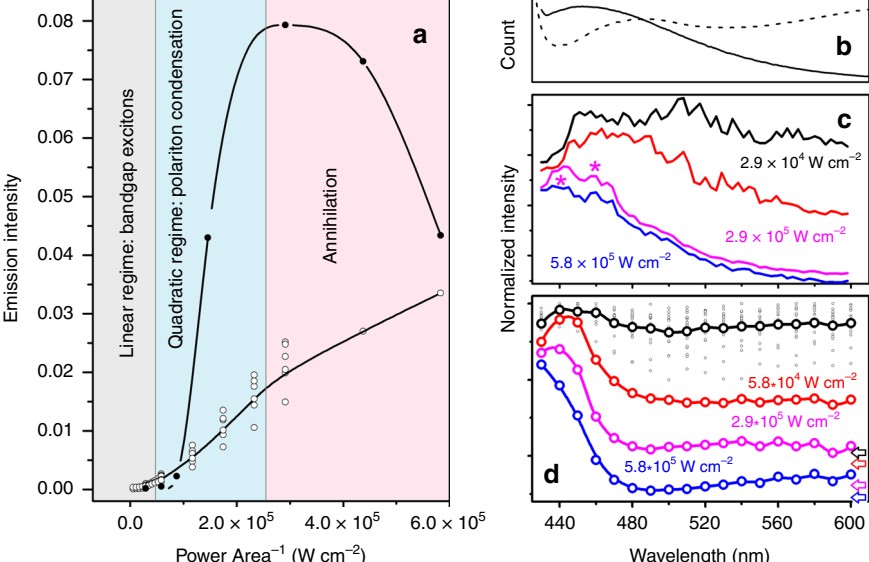

**Fig. 2 Emission intensity and spectral properties in dependence on power. a** Emission intensity dependence on fluence of excitation radiation at 405 nm with emission detection at 430 nm for a ZnO microcrystal (open circled line) and for a ca. 2 µm ZnO microsphere (closed circle line): the points present experimental data and the curves are to guide the eye. **b** Selected from Fig. 1 steady-state emission spectra (405 nm excitation) for a ZnO microcrystal (dashed line) and for a ca. 2 µm ZnO microsphere (solid line). **c** Emission spectra (normalized on maxima) for a ZnO microcrystal in dependence on fluence of excitation radiation at 405 nm. Stars indicate spectral signatures which could be signatures of whispering gallery like modes or localized cavity excitations. **d** Emission spectra (normalized on maxima) detected for a ca. 2 µm ZnO microsphere in dependence on fluence of excitation radiation at 405 nm. Spectra in the panels (**c**, **d**) are normalized and, for clarity, shifted vertically: the corresponding color arrows indicate the zero level for each spectrum.

## Discussion

Nonlinear amplification of blue shifting emission was reported in micro-cavities[1,21,22] as a signature of a nonlinear parametric process to condense polaritons and to populate an exciton blue-shifted reservoir[23,24]. The data reported in Fig. 2 provide strong evidence that in relatively large quasi-spherical polycrystalline microcrystals and in relatively small polycrystalline microspheres, there is a common physics of coupling of the external field to the material response. This is consistent with the fact that under a spherical boundary, a material with a relatively high refractive index may stabilize internal field modes and provide amplification such as in stable cavities[18,19]. Furthermore, properties of emission on power increase show when we reach the critical concentration needed to start polariton BE condensation. Above $2 \times 10^5$ W cm$^{-2}$, emission efficiency in the larger microcrystals demonstrates a semi-linear tendency. In the same power regime, emission from the microspheres demonstrates a significant attenuation of efficiency. We ascribe this to stronger confinement for the field component in the microspheres, which are smaller and more ideal as spherical structures. It is important to note that at the high-power limit, we do not observe spectral signatures reported for electron-hole plasma in ZnO microstructures[32,33]. This suggests that under higher fluence, exciton–exciton annihilation[34] may become dominant in the dynamics of excitons. An alternative perspective would ascribe the observed attenuation of the polariton nonlinear regime to the effect of the physical boundary of the ZnO cavity that can no longer confine the photonic part of the wave-function of the polariton ensemble, which is expected to expand upon higher excitation[35].

Formation of quantum droplets implies delocalizing tendencies, which can be addressed using confocal microscopy sampling. As the field confinement effects are more prominent in microspheres, we describe the behaviour of this system here. To explore analogous data for the larger microcrystal, please refer to Supplementary Note 9. In Fig. 3a, b we present images of the band gap excitations sampled at 430 and 530 nm for ZnO

microspheres. Consistently, in Fig. 3c, d we show their Fourier transform (FT) power spectra in momentum space of spatial frequencies. Even though the fine details of possible WG modes as anticipated for the ZnO microspheres studied are beyond the resolving capacity of the microscope, the periodic patterns in the outer spectral rings in Fig. 3c confirm the presence of cavity modes in the microspheres: here, we refer to computed radial patterns in momentum space we present in the Supplementary Note 10. Furthermore, in Fig. 3e–h we show changes in Fourier transformed spatial frequency power spectra when we increase the power of the excitation field from 10 to 50% (from $5.8 \times 10^4$ to $2.9 \times 10^5$ W cm$^{-2}$), and from 50 to 100% (from $2.9 \times 10^5$ to $5.8 \times 10^5$ W cm$^{-2}$), respectively. Before taking the differences (as shown), we normalized the spatial frequency power spectra by the sum of their readings. For images taken at 430 nm, the system demonstrates dominant peaks centred at the zero spatial frequency (see panels (e) and (f)). These are accompanied by depletion of contributions of higher spatial frequencies. For example, the spatial spectrum in Fig. 3e the negative (blue) component becomes prominent at spatial frequencies larger than $7.6 \times 10^5$ m$^{-1}$. The results imply significant electronic delocalization. For images taken at 530 nm (see Fig. 3f) the same delocalization component is present, but its contribution is smaller. This suggests that localization dominates in power-dependent spatial redistributions of the states emitting at 530 nm. In the higher range of power change, the ZnO microsphere demonstrates a significant decrease of the peak at the zero frequency for the image taken at 430 nm and a moderate decrease of the peak at the zero frequency for the image taken at 530 nm, see Fig. 3g, h, respectively. Comparing with the results shown in Fig. 2a, we may state that in the microsphere the efficiency decay at higher power (fluence) is associated with spatial localizations.

The power-dependent changes in the FT spatial spectra indicate that relaxations experience different pathways that compete in dependence on the excitation power used. To describe this, we use spectral differences shown in Fig. 3c and 3d to construct

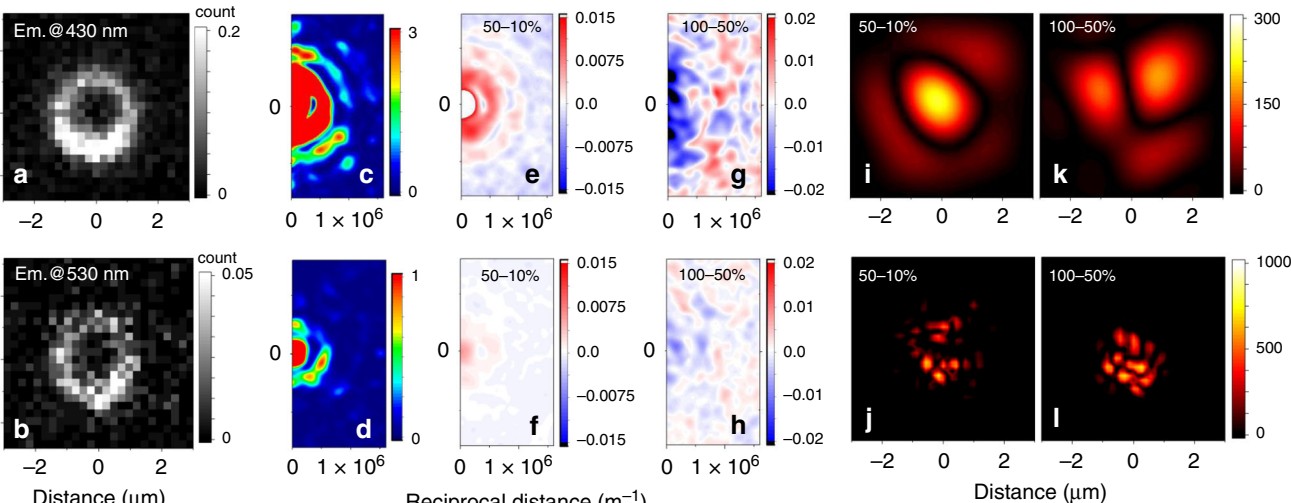

**Fig. 3 Fourier filtered power-dependent redistribution of emission spatial components. a, b** Confocal emission scans for ca. 2 μm ZnO microspheres detected at 430 and 530 nm, respectively, using 405 nm for excitation. **c, d** Spatial frequency spectra (by Fourier Transform of images into momentum space) of images shown in panels (**a, b**). **e, f** Differences of normalized 2D spatial frequency spectra (by Fourier Transform of images) of images taken on 2 μm ZnO using 50% ($2.9 \times 10^5$ W cm$^{-2}$) and 10% ($5.8 \times 10^4$ W cm$^{-2}$) of laser power. **g, h** Analogous differences of normalized 2D spatial frequency spectra while using 100% ($5.8 \times 10^5$ W cm$^{-2}$) and 50% ($2.9 \times 10^5$ W cm$^{-2}$) of laser power. Normalizations were conducted taking sums of intensities of all pixels. A 2D spatial spectrum is a radial function: here, we position zero frequency in the middle of the left vertical axis. **i, j** Inverse FT after application of a low-frequency path filter (from 0 to $2 \times 10^5$ m$^{-1}$) to the difference of emission images detected at 430 nm or 530 nm upon excitation power increase from 10 to 50%. **k, l** Inverse FT after application of a band-pass filter (from $1.3 \times 10^5$ to $9.8 \times 10^5$ m$^{-1}$) to the difference of emission images detected at 430 nm or 530 nm upon excitation power increase from 50 to 100%.

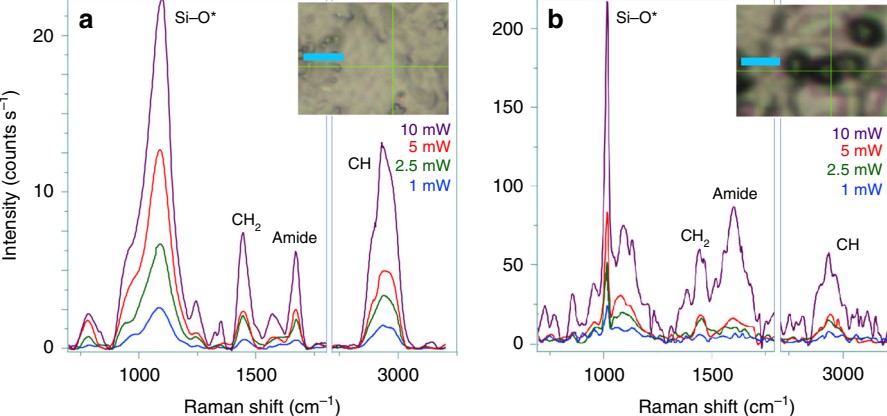

**Fig. 4 Demonstration of surface polariton enhanced Raman. a** Raman spectra of G12 polypeptide deposited on a borosilicate glass microscope slide in dependence on power of radiation at 405 nm, as indicated, using ×100 (0.8 NA, 3.4 mm working distance) objective. **b** Raman spectra for the polypeptide deposited on ZnO microspheres on a borosilicate glass microscope slide under the same excitation powers as in panel (**a**). Insets in each panel show bright field microscopy images where Raman spectra were taken: cyan bars indicate a 5- μm length.

low-frequency ($0$ to $2 \times 10^5 \, \mathrm{m}^{-1}$) and intermediate-frequency ($1.3 \times 10^5$ to $9.8 \times 10^5 \, \mathrm{m}^{-1}$) two-dimensional spectral filters. Using the filters, we separate spectral components of normalized images detected under different powers. Next, we take the inverse Fourier transform (IFT) to map separated spectral components in the image plane. For example, Fig. 3i, j shows how spatial distribution (in the image plane) of the states emitting at 430 nm (which contribute into the low-frequency spectral region in Fig. 3e, g) would change upon power increase. Specifically, Fig. 3i describes the delocalizing tendency upon excitation power increase from 10 to 50% when the efficiency power dependence (as shown in Fig. 2a) is quadratic. We may suggest that the image describes the luminous part of the field of the interacting polaritons, which is expected to leak outside the material boundary[36]. Figure 3k clearly indicates that when we use an excitation power above the quadratic regime, the low-frequency filtered difference spatial components demonstrate nodal or discontinuing segmentation across the structure in the image plane. Analogously, in Fig. 3j, l we present the results of application of the intermediate-frequency filter: we see that upon power increase in the quadratic regime, the excited carriers tend to shift towards the surface: Fig. 3j. If under the higher fluence, the carriers start to fill the internal part of the ZnO structure: Fig. 3l.

What is the nature of the localizing dynamics? Since Oxygen vacancies were anticipated to be 0.5–0.8 eV above the valence-band maximum[37] and Zn vacancies calculated to be 0.1–0.2 and 0.9–1.2 eV above the valence-band maximum, respectively[38], then both types of defect are expected to contribute to the luminescence. These frequencies are significantly lower that the band-gap emission: for example, the band centred around 2.4 eV[39]. We ascribe the observed localization of some of the carriers as shown in Fig. 3j, l, to diffusive trapping on Zn interstitials and shallow donors distributed in the volume of the ZnO structures[38,40]. The anticipated trapping dynamics may facilitate the exciton–exciton annihilation mechanism[34] which would explain the emission attenuation under higher excitation fluence, see Fig. 2a. Power-dependent FT spatial spectroscopy reports, however, that the delocalizing dynamics of polariton droplets compete with localizing tendencies due to the granulated polycrystalline nature and to the contribution of diffusive trapping on Zn interstitials. The latter would provide correlations and dynamics specific to Anderson media[41]. Under the high-power regime, multi-particle annihilation and, consequently, ionization become competitive.

Stimulation of the field component of polaritons (see Fig. 3i, k) to expand beyond the physical boundary of ZnO may offer field enhancement at the surface. In contrast to the surface plasmon of collective oscillations of conduction electrons in metals[42], the considered mechanism would borrow from the leaky field part of the interacting polaritons. To test this, we explored the interaction of the zinc oxide specific G12 polypeptide[25]. In Fig. 4a, b we compare power-dependent Raman responses of the polypeptide, when deposited on a glass microscopy slide alone and on the top of ZnO microspheres, respectively. While for the former system, Raman responses are linear in dependence on power, for the latter, when the power is increased from 5 to 10 mW, the amide and the CH modes of the polypeptide at ZnO demonstrate a 10-time intensity enhancement. In Supplementary Note 11 we further compare Raman responses of the system using excitation at 405 and 532 nm. It is interesting to observe that Si–O$^-$ stretching vibrations in $Q_3$-structural elements[26,27] in the borosilicate glass (D263M) substrate demonstrate a nonlinear increase in intensity when next to microspheres of ZnO. The enhanced narrow spectral response at 1010 cm$^{-1}$ is relatively red shifted compared to the main $Q_3$ band at 1100 cm$^{-1}$, as shown in Fig. 4a. This suggests a high degree of selectivity in boosting polarization of certain Si–O modes in the contact point between flat glass and the ZnO microsphere. Consistently, relatively broad widths of the enhanced responses from polypeptide moieties are due to structural variances and orientational distributions of the polypeptides deposited on the microsphere. The apparent selectivity in enhancement of Si–O modes in contact with ZnO suggests that the reported SPFE has a capacity to complement interface specific sum-frequency generation spectroscopy[43].

## Methods

**Synthesis and material studies**. Zinc oxide microcrystals (average size of 100 μm) are the analytical standard from Sigma-Aldrich, CAS Number 1314-13-2. We synthesize ZnO microspheres and G12 polypeptide as reported previously[20,25]: for details see Supplementary Note 1. Using PANalytical X'Pert PRO X-ray diffractometer (Malvern Panalytical, Malvern, UK) with Cu Kα radiation at 1.54056 Å we extract the crystallite domain sizes of 52 ± 9 and 18 ± 5 nm for microcrystals and microspheres, respectively[44].

**Emission spectroscopy and microscopy**. Steady-state two-dimensional emission spectra[28] of solid-state samples are assembled using conventional emission spectra taken with TECAN i-control M200 spectrometer (Tecan Group Ltd., Switzerland) and well plates Costar (Corning, US). We conduct confocal emission microscopy with Leica TCS SP5 station (Leica, Milton Keynes, UK) equipped with ×63 objective (0.5 numerical aperture). Under the rate of 400 MHz, we scan a spatial region of 124 × 124 μm$^2$ with resolution of 0.242 μm in both directions of the image plane. We provide further details on the confocal microscopy scanning regime in Note 1 of the Supplementary Information.

**Raman microscopy**. Raman spectra were obtained using a LabRAM HR Evolution confocal Raman microscope (HORIBA UK Ltd at Northampton) with a ×100 (0.8 NA, 3.4 mm working distance) and excitation at 405 nm.

**Numerical simulations**. Simulations of the electric field properties in spherical cavities in dependence on radius and polarization of excitation were conducted using a FDTD approach implemented by Ansys/Lumerical Inc. The same package was used to calculate Q-factors according to an approach reported elsewhere[45]. We provide a detailed account of numerical studies in Note 3 of the Supplementary Information.

## Data availability
The data presented in this study is available from the authors on reasonable request.

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

## Acknowledgements
Funding from AFOSR FA9550-1-16-2013 is gratefully acknowledged. The authors are grateful to Dr. Adam Holland and Dr. David Sheppard for assistance in Raman microscopy studies using the research infrastructure of HORIBA UK Ltd at Northampton, United Kingdom.

## Author contributions
All authors contributed to the design of the experiments and to the collection and analysis of data. C.C.P. performed synthesis of ZnO microspheres. D.J.O. and V.V.V. conducted emission experiments. V.V.V. and A.H. conducted Raman studies. V.V.V. analyzed the emission microscopy results and conducted the numerical simulations. V.V.V. and C.C.P. analyzed the data and wrote the manuscript. The final version of the manuscript has been approved by all authors.

## Competing interests
The authors declare no competing interests.
