## [Peer Review File · Nature Communications]

Reviewers' comments:

Reviewer #1 (Remarks to the Author):

The authors report on polariton emissions in ZnO microstructures and claim that the observed polariton related surface field can be used for enhancing Raman signals. This may be one the new applications of exciton polariton, however, the manuscript needed to be rewritten and the following points need to be addressed carefully to reach the criteria of NC.

1. How do the authors prove the emissions are in fact from polaritons?
2. How are the confocal luminescence scanning measurements carried out? The authors mention that the scan is at equatorial plan, how is this done? As the focal points inside the ZnO microcrystals are different from those at the air-ZnO surface.
3. Can the authors numerically calculate the field distribution in microcrystals? I suggest that this should be done and compare with the experimental results.
4. In Figure C, the authors claim that they observe the whispering gallery modes, to me, these are in the same order of noise. If these are the WG modes, the numerical calculations can give rough ideas of the mode order, Q-factors etc.
5. What's the advantage of the surface polariton enhanced Raman? Compared to surface plasma enhancement (several orders stronger), I can't really see the obvious advantage.
6. Several technical issues: (1) add color scales to the contour maps (Fig. 1 and 3); (2) In Fig. 2A, please indicate that the curves are in fact eye guide. (3). In Fig. 1D1, the scale bar's value size is too small. Fig. 1D2, no scale bar length unit is shown.

Reviewer #2 (Remarks to the Author):

In this manuscript "Beyond the cavity: polariton confinement and condensation in spherical ZnO micro-crystals: implications for surface enhanced Raman" authors have reported 3D polariton confinement and condensation in pseudo-spherical ZnO microcrystals and studied field enhancement behaviour of polaritons at the surface. The reported surface polariton field enhancement of Si-O modes in contact with ZnO (the interaction of a polypeptide) suggests that ZnO has a capacity to complement interface specific sum-frequency generation spectroscopy. Confined polaritons in ZnO have already been extensively investigated in one-dimensional (1D) or two-dimensional (2D) microcavities, however, three-dimensional (3D) confinement of polariton, as well as its applications, has rarely been explored. Also, the realization of three-dimensionally confined polariton states in ZnO is still a challenging task. In this work, the authors have made an interesting work which opens up opportunities for research and engineering of bio-inorganic and inorganic composites using zinc oxide as both a structural and sensing element. The manuscript can be considered for publication in this journal after addressing the following issues.

1. What are the factors affecting the confinement of polaritons in the ZnO microsphere. What is the direction of momentum vector in the cross-sectional plane?
2. Try to include the graph showing periodic oscillation of intensity with 'k', in order to confirm the 3D confinement of polaritons
3. Calculate the fitness of the cavity. Since, larger fitness value indicates high quality of the cavity and stronger confinement.
4. If possible record Raman with a slightly different excitation (example 514 nm) and see differences if any.

Sd/-

Dr K. G. Gopchandran

Professor

Department of Optoelectronics

University of Kerala

Kariavattom

Thiruvananthapuram-695581, India.

Reviewer #1:

The authors report on polariton emissions in ZnO microstructures and claim that the observed polariton related surface field can be used for enhancing Raman signals. This may be one the new applications of exciton polariton, however, the manuscript needed to be rewritten and the following points need to be addressed carefully to reach the criteria of NC.

Reply: General response to the narrative above.

Indeed, even though polariton emission in ZnO (in bulk) is always present, usually, its contributions is small if not negligible comparing to other relaxation pathways. In our contribution, we provide experimental evidences (and now, according to your suggestions, they are supported by simulations), for a significant role of polariton dynamics in micro-spherical ZnO. This is due to the microspherical structuring and the excitation regime, which allow sufficient energy density stored in cavity modes to interact with material response. Furthermore, we demonstrate that under intense excitation, collective polariton phenomena can be observed.

Second, it is not the purpose of our study to search for possible Raman enhancement effects to make it competitive to SERS at metal surfaces in various analytical diagnostics. We submitted our manuscript to Nature Communication because we observe a novel type of Raman enhancement which is not like SERS. It may never be competitive to SERS at metal surfaces. The very purpose of our contribution is to show readers that the physics present in the observed phenomenon is different. Polariton dynamics is not the same as plasma dynamics and polariton interactions are under rules of quantum correlations which are not the same for an electronic gas. In the following replies we defend our perspective. We took your narrative and the questions you pose as very valuable instructions, and we clarified the text. Thank you.

1. How do the authors prove the emissions are in fact from polaritons?

Answer: To answer this first let us account that the polariton regime was anticipated theoretically in the early 50s [K. Tolpygo. J. Exp. Theor. Phys. 20, 497-509 (1950); H. Kun. Lattice vibrations and optical waves in ionic crystals. Nature 167, 779–780 (1951); A. Davydov. Theory of Solids. Nauka 1976, Chapter 3] was observed in ZnO as a minor process at 10K. As the studied system did not provide

A comprehensive review of ZnO materials and devices

Cite as: J. Appl. Phys. 98, 041301 (2005); <https://doi.org/10.1063/1.1992666>
Submitted: 02 February 2005 . Accepted: 13 June 2005 . Published Online: 30 August 2005

Ü. Özgür, Ya. I. Alilov, C. Liu, A. Teke, M. A. Reshchikov, S. Doğan, V. Avrutin, S.-J. Cho, and H. Morkoç

FIG. 52. Bound-excitonic region of the 10-K PL spectrum for a forming-gas-annealed ZnO substrate. [Phys. Rev. B 70, 195207 (2004).]

FIG. 50. Free-excitonic fine-structure region of the 10-K PL spectrum for a forming-gas-annealed ZnO substrate. [Phys. Rev. B 70, 195207 (2004).]

In strongly polar materials such as ZnO, transverse Γ_3 excitons couple with photons to form polaritons. In principle, although the polaritons can be formed anywhere along the dispersion curves, polariton lifetimes, which are higher at certain points, determine the observed peak positions. Therefore, as indicated in Fig. 50 the $FX_A^{n=1}(\Gamma_3)$ exciton line has two components. The higher energy component at 3.3810 eV, which is 3.6 meV apart from the A exciton, is assigned to the so-called longitudinal exciton (upper polariton branch, UPB_A). The lower energy component at 3.3742 eV, which is about 2.9 meV apart from the A exciton, corresponds to the recombination from the “bottleneck” region, in which the photon and free-exciton dispersion curves cross (lower polariton branch, LPB_A).

Reprinted figures with permission from the American Physical Society as follows: Teke, et al., Excitonic fine structure and recombination dynamics in single-crystalline ZnO, *Phys. Rev. B* 70, 195207 (2004). Copyright 2020 by the American Physical Society. <https://doi.org/10.1103/PhysRevB.70.195207>

Reprinted figures from Özgür, et al., A comprehensive review of ZnO materials and devices. *Journal of Applied Physics* 98, 041301 (2005), with the permission of AIP Publishing. <https://doi.org/10.1063/1.1992666>

effective coupling between light and matter: the reported Rabi splitting was small and the relative contributions of the upper and the lower branches were small. To stimulate polariton dynamics at room temperature, we need to provide significant field density to co-exist with excitons, and excitons should be stable-coherent to interact with the field effectively.

In our studies, first, we take advantage of the fact that a spherical cavity would have the highest possible quality factor (30%-40% higher than that of a flat one) [C. A. Balanis. *Advanced Engineering Electromagnetics*, Wiley, 1989, p. 566]. Therefore, we explore comparatively properties of relatively large semi-spherical microcrystals and relatively high quality microspheres, with radii that vary between 1 and 2 micron. Here, we wish to notice that the spectral heterogeneity at the red side of the exciton peak in the emission excitation spectra measured in ensembles of microstructures (see Figures 1C₁ and 1C₂) may include contribution of the Lower Polariton branch if present. In the case of microspheres, the spectral heterogeneity at the red side of the exciton peak is more prominent. Of course, such experimental observation alone would not be sufficient to predict that the considered microstructures may provide effective coupling of field with material response.

To progress, second, we borrowed from knowledge reported on the properties of polariton dynamics in cavities [Nature 443, 409 (2006), New J. Phys. 14, 013037 (2012), Nature 418, 751 (2002); Phys. Rev. B 32, 6601–6609 (1985), Phys. Rev. B 61, 13856 (2000); Nat. Com. 9, 2944 (2018)]. Accordingly, second, we conducted emission micro-spectroscopy on selected microstructures using a tight focusing excitation regime in dependence on power. Specifically, we used fluences, which were reported to provide the condensation regime [Sov. Phys. Solid State 6, 2219 (1965), Sov. Physics JETP 27, 521 (1968), New J. Phys. 14, 013037 (2012)]; and we excited at the red side to avoid ambiguities due to possible bi-stabilities [Sov. Phys. JETP 36, 767 (1973)]. Additionally, in our studies we used a high rate injection regime (400 MHz) provided by the confocal microscope. Approaching fluences known to induce condensation, we did observe the nonlinearity of emission efficiency (Figure 2A), the blue shift (Figures 2D and 2C) and the spatial delocalization of emission (Figure 3). These features were reported as signatures of polariton dynamics and condensation in cavities [Nature 443, 409 (2006), New J. Phys. 14, 013037 (2012), Nature 418, 751 (2002); Phys. Rev. B 32, 6601–6609 (1985), Phys. Rev. B 61, 13856 (2000); Nat. Com. 9, 2944 (2018)]. Here, we would like to note that according to the spectral window of the microscope, we observe the peak of emission to shift from 450 to 420 (and higher). The spectral range of the blue shift covers a half of the spectral range of the heterogeneous shoulder at the red side of the exciton peak (Figure 1C₂),

Considering the observations, we rule out super-radiance phenomena based on exciton (not polariton) physics. The observations suggest that the spectrally heterogeneous shoulder at the red side of the exciton peak (Figure 1C₂) does include Rabi resonances of frequencies that vary according to variance of light-matter coupling in various microspheres. The analogous red side shoulder in the excitation spectrum in microcrystals is significantly lower (Figure 1C₁). This is consistent with the fact that stimulation of polariton dynamics and correlations in micro-crystals is barely possible (see Figures 2A and 2D). However, it is important to state here that the fact that we observe analogous tendencies (but to a different extent) in spherical or pseudo-spherical microstructures of different sizes provides further support that we managed to enhance the same polariton regime [K. Tolpygo. J. Exp. Theor. Phys. 20, 497-509 (1950); H. Kun. Lattice vibrations and optical waves in ionic crystals. Nature 167, 779–780 (1951); A. Davydov. Theory of Solids. Nauka 1976, Chapter 3].

To conclude here, we wish to notice that according to your suggestions we conducted theoretical studies which confirm that micro-spherical structures under intense excitation (as we described) would provide cavity modes to stimulate the polariton regime in the spectral region explored.

2. How are the confocal luminescence scanning measurements carried out? The authors mention that the scan is at equatorial plan, how is this done? As the focal points inside the ZnO microcrystals are different from those at the air-ZnO surface.

Answer: We provide a detailed account of the experiment in the second section of the Supporting information file, Indeed, while working with small microspheres it may not be easy to secure that we are scanning in the equatorial plane. To manage this in the confocal microscopy experiments we program the microscope to collect data at different heights and sorted suitable images afterwards.

3. Can the authors numerically calculate the field distribution in microcrystals?

Answer: We conducted numerical simulations: please, see third section in the updated Supporting Information file: for example, please see Figures S6, S8-S10, S14-S16

We agree with you that our declaration in the main text that we observed WG modes in large microcrystals was not careful. We updated the text taking account of the results of the simulations: In particular, we softened our definition saying that the spectral signatures in the large microcrystals are due to WG-like or more localised modes. In the case of the smaller microspheres, theory suggests the spectral widths of the WG modes (see Figures 12) to be comparable with the blue shifting spectral component which is under nonlinear enhancement. At the same time, this component is still at the red side of the bulk bandgap - in the spectral region of the polariton Rabi resonance. Under intense excitation of cavity modes, we stimulate polariton dynamics in the small microspheres.

4. I suggest that this should be done and compare with the experimental results.

Answer: There is a sub-chapter in the third section in the updated Supporting Information file with Figure S11. Accordingly, in the sub-section, we discuss simulated results to match the grid according to spatial resolution as present in experiment. The simulated pattern confirms the experimental observations of intensity increase at the edges (rim like patterns) in the emission images of the small microspheres.

Here, we wish to notice that Lumerical simulations provide field distributions only. This is important to predict where and how polariton density would be distributed. However, the simulations do not account the exact properties of polariton wavefunctions and how such would change under different excitations [PRB 2012, 86: 195305]. Because of this, in the main text, we are rather careful in our correlation of experiment and simulations by Lumerical. We are experts in materials science, in inorganic synthesis and in experimental spectroscopy first of all. The most important results are experimental and they are shown in Figures 2-4.

5. In Figure C, the authors claim that they observe the whispering gallery modes, to me, these are in the same order of noise. If these are the WG modes, the numerical calculations can give rough ideas of the mode order, Q-factors etc.

Answer: First of all, the spectral structure we observe is not due to noise: note, we normalized on the maximum. The same Figure provides the nonlinearity dependence on intensity. If you scale up, you would see that the spectral signature we assigned to WG are much more intense than the noise we report for the spectral average detected under lower power.

Second, we agree that our original definition was not careful. As we noticed before, we updated the text taking account of the results of the simulations: we softened the definition saying that the spectral signatures in large microcrystals are due to WG-like or localised modes.

We calculated the Q-factors for microspheres: please, see Figure S13 in the Supporting Information file. The factors are in the range of the values reported for high quality cavities to stimulate condensation [C. R. Physique 17, 946 (2016)].

6. What's the advantage of the surface polariton enhanced Raman? Compared to surface plasma enhancement (several orders stronger), I can't really see the obvious advantage.

Answer: Considering Raman enhancement provided by silver nanoparticles, the observed nonlinearity at the surface of ZnO microspheres is not "competitive". However, searching for advantages was not our goal. In the manuscript, first of all, we report on a novel physics of Raman

enhancement, the nature of which is yet to be explored in dependence on material properties, excitation and relaxation dynamic regimes. Polariton dynamics is expected to compete with relaxation, trapping and ionization. At the same time, polariton dynamics is known to demonstrate peculiar properties due to quantum droplet formation and possible correlations in vortices. Linear and nonlinear responses of such systems would have to be reconsidered from the perspective of quantum dynamic (forming and dissipating) of such collective states.

Having an advantage and making a use of something are possible when we understand details. We report on a new observation, that the wide research community was not aware of before. Analogously, when the first magnetic echo signal was reported, the authors did not think that it could lead to NMR spectroscopy.

7. Several technical issues: (1) add color scales to the contour maps (Fig. 1 and 3); (2) In Fig. 2A, please indicate that the curves are in fact eye guide. (3). In Fig. 1D1, the scale bar's value size is too small. Fig. 1D2, no scale bar length unit is shown.

Answer: Color scales are added in Fig 1 and 3. Now, in the caption to Fig. 2A, we indicate that the curves act as eye guides. We increased size of the scale bar and of the unit notation in Figure 1D₁, and we rewrote the figure caption to make the description clearer. We provide the length unit for the scale bar in Figure 1D₂ and edited the figure caption for better clarity. Thank you.

Reviewer #2 (Remarks to the Author):

In this manuscript “Beyond the cavity: polariton confinement and condensation in spherical ZnO micro-crystals: implications for surface enhanced Raman” authors have reported 3D polariton confinement and condensation in pseudo-spherical ZnO microcrystals and studied field enhancement behaviour of polaritons at the surface. The reported surface polariton field enhancement of Si-O modes in contact with ZnO (the interaction of a polypeptide) suggests that ZnO has a capacity to complement interface specific sum-frequency generation spectroscopy. Confined polaritons in ZnO have already been extensively investigated in one-dimensional (1D) or two-dimensional (2D) microcavities, however, three-dimensional (3D) confinement of polariton, as well as its applications, has rarely been explored. Also, the realization of three-dimensionally confined polariton states in ZnO is still a challenging task. In this work, the authors have made an interesting work which opens up opportunities for research and engineering of bio-inorganic and inorganic composites using zinc oxide as both a structural and sensing element. The manuscript can be considered for publication in this journal after addressing the following issues.

1. What are the factors affecting the confinement of polaritons in the ZnO microsphere. What is the direction of momentum vector in the cross-sectional plane? Try to include the graph showing periodic oscillation of intensity with ‘k’, in order to confirm the 3D confinement of polaritons

Answer: In the manuscript, first of all, we focus on how size reduction affects nonlinearity of the band-gap emission. Specifically, how the size of ZnO particles would facilitate the initial part of the polariton condensation process and how this, when at higher fluences, would be overshadowed by scattering, ionization or special redistributions of the photon and exciton components of polaritons [Phys. Rev. B 86, 195305 (2012)]. We keep in mind a “generic” quantum phase diagram: for your convenience, here, we present several, which we collected from different sources as indicated.

[Proc. SPIE 10102, Ultrafast Phenomena and Nanophotonics XXI, 101020T (23 February 2017); doi.org/10.1117/12.2250735; Advances in Condensed Matter Physics 2011 Article ID 938609; doi.org/10.1155/2011/938609].

Content reprinted by permission from SPIE: Sekiguchi, et al., Exciton Mott transition in GaAs studied by terahertz spectroscopy, Proc. SPIE 10102, Ultrafast Phenomena and Nanophotonics XXI, 101020T (23 February 2017)

The content reported by Hindwai is reproduced here under an Attribution 3.0 Unported (CC BY 3.0) licence; <https://creativecommons.org/licenses/by/3.0/>

Polariton physics accounts for a correlation between the material response and the field distribution.

Your request concerns distribution of field alone. Indeed, this is very important as it is the first step towards the rigorous theory of polaritons in micro-structures. To answer, we conduct FDTD theory simulations using the Lumerical package: please see the results in the third section of the updated Supporting Information.

According to FDTD theory, a micro-spherical cavity of a smaller radius would support a smaller number of modes which would tend to have higher intensity at the lower wavelength side, and would tend to become spectrally broader. The predictions are in agreement with the tendencies one can observe using adjustable Fabry-Perot. FDTD simulations in disks of radii larger than 10 micron (the

data are not provided as weakly relevant) indicate that numerous and spectrally narrow cavity modes start to form unresolved bands. If we consider spatial properties, a mode of the same frequency but in a smaller microsphere would be spatially broader along the radial dimension. Simulations in disks of radii larger than 10 micron in an excited field tend to fail to form a standing cavity mode, but would tend to localise along proximal curvature, unless excitations were seeded densely: the data are not provided as weakly relevant to the current contribution.

Are the calculated results relevant to the effect of confinement as we bring under attention?

Yes, they are.

Cavity modes are orthogonal: the more modes a cavity would have in a spectral range of interest, the more ways we may store energy in that spectral range in the cavity without destructive interference. FDTD simulations suggest that a smaller microsphere provides a lower number of modes to be filled in the smaller space and with a lesser spatial certainty. Therefore, in such systems, under increasing fluence we may expect enhancement of collective (constructive or destructive) phenomena: presenting emission nonlinearity, its collapse and the spectral blue shift in dependence on power and size, as shown Figure 2 in the Main text, we try to articulate this aspect.

To answer your critical concern, we updated the Supporting Information file with the results of FDTD simulations as you suggested and edited the Main text.

3. Calculate the fitness of the cavity. Since, larger fitness value indicates high quality of the cavity and stronger confinement.

Answer: We agree with you that a higher quality cavity would store energy better and longer. Furthermore, efficiency of energy storing for modes of different frequencies (supported by the same cavity) may be different. Indeed, within the context of engineering of cavity photonics, energy storing may be described as “confinement”. However, in the article we discuss the effects of size reduction on multi-body phenomena in respect to polariton dynamics: please, see Figure 2 and the discussion in the main text.

Nonetheless, the critical concern you raised is very important!

Accordingly, using FDTD theory, we managed to calculate Q-factors for microspheres of radii smaller than 2 micron: please, see Figure S13 in the Supporting Information file. The factors are in the range of the values reported for high quality cavities to stimulate polariton condensation [C. R. Physique 17, 946 (2016)].

4. If possible record Raman with a slightly different excitation (example 514 nm) and see differences if any.

Answer: We update the Supporting Information file with Figure S19 where we compare Raman spectra detected using 405 and 530 nm, and comment on the observed differences.

REVIEWERS' COMMENTS:

Reviewer #1 (Remarks to the Author):

The authors have revised their manuscript carefully and have addressed all my concerns. I don't have further questions. It is much better than the previous version and I believe that it reach the criteria of NC.

Reviewer #2 (Remarks to the Author):

The authors addressed my comments satisfactorily, in the revised manuscript and I am happy to recommend the manuscript for publication.

Response to Referees

REVIEWERS' COMMENTS:

Reviewer #1 (Remarks to the Author):

The authors have revised their manuscript carefully and have addressed all my concerns. I don't have further questions. It is much better than the previous version and I believe that it reach the criteria of NC.

Reviewer #2 (Remarks to the Author):

The authors addressed my comments satisfactorily, in the revised manuscript and I am happy to recommend the manuscript for publication.

No comments to address